# Mitochondrial DNA Variants in Patients with Liver Injury Due to Anti-Tuberculosis Drugs

**DOI:** 10.3390/jcm8081207

**Published:** 2019-08-13

**Authors:** Li-Na Lee, Chun-Ta Huang, Chia-Lin Hsu, Hsiu-Ching Chang, I-Shiow Jan, Jia-Luen Liu, Jin-Chuan Sheu, Jann-Tay Wang, Wei-Lun Liu, Huei-Shu Wu, Ching-Nien Chang, Jann-Yuan Wang

**Affiliations:** 1Department of Laboratory Medicine, Fu Jen Catholic University Hospital, Fu Jen Catholic University, New Taipei City 24352, Taiwan; 2Department of Internal Medicine, National Taiwan University Hospital, Taipei 10002, Taiwan; 3Department of Laboratory Medicine, National Taiwan University Hospital, Taipei 10002, Taiwan; 4One-Star Technology, New Taipei City 24352, Taiwan; 5Foundation of Liver Diseases, Taipei 10002, Taiwan; 6School of Medicine, College of Medicine, Fu Jen Catholic University, and Department of Emergency and Critical Care Medicine, Fu Jen Catholic University Hospital, Fu Jen Catholic University, New Taipei City 24205, Taiwan; 7Department of Surgery, Fu Jen Catholic University Hospital, Fu Jen Catholic University, New Taipei City 24352, Taiwan

**Keywords:** Drug-induced liver injury, tuberculosis, mitochondria, DNA variants, complex I

## Abstract

Background: Hepatotoxicity is the most severe adverse effect of anti-tuberculosis therapy. Isoniazid’s metabolite hydrazine is a mitochondrial complex II inhibitor. We hypothesized that mitochondrial DNA variants are risk factors for drug-induced liver injury (DILI) due to isoniazid, rifampicin or pyrazinamide. Methods: We obtained peripheral blood from tuberculosis (TB) patients before anti-TB therapy. A total of 38 patients developed DILI due to anti-TB drugs. We selected 38 patients with TB but without DILI as controls. Next-generation sequencing detected point mutations in the mitochondrial DNA genome. DILI was defined as ALT ≥5 times the upper limit of normal (ULN), or ALT ≥3 times the ULN with total bilirubin ≥2 times the ULN. Results: In 38 patients with DILI, the causative drug was isoniazid in eight, rifampicin in 14 and pyrazinamide in 16. Patients with isoniazid-induced liver injury had more variants in complex I’s NADH subunit 5 and 1 genes, more nonsynonymous mutations in NADH subunit 5, and a higher ratio of nonsynonymous to total substitutions. Patients with rifampicin- or pyrazinamide-induced liver injury had no association with mitochondrial DNA variants. Conclusions: Variants in complex I’s subunit 1 and 5 genes might affect respiratory chain function and predispose isoniazid-induced liver injury when exposed to hydrazine, a metabolite of isoniazid and a complex II inhibitor.

**Take home message:** Mitochondrial DNA variants in *ND1* and *ND5* genes (coding NADH dehydrogenase subunit 1 and 5 of complex I) might affect the oxidative phosphorylation function of complex I and are associated with the risk of drug-induced liver injury (DILI) due to anti-TB drugs, especially isoniazid. 

## 1. Introduction

Hepatotoxicity is the most severe adverse effect of antituberculosis (anti-TB) therapy [1]. Hepatotoxicity may lead to liver failure that is fatal or result in a need for liver transplantation [2]. The mechanism of hepatotoxicity due to anti-TB drugs remains unclear, although toxic metabolites such as hydrazine, a metabolite of isoniazid, have been suggested to play a crucial role [3,4]. Mitochondrial dysfunction is an essential mechanism in drug-induced liver injury (DILI) due to acetaminophen, amiodarone, diclofenac, propofol, valproic acid and many others [5,6]. The dysfunction can be caused by abnormalities in mitochondrial membrane permeabilization, oxidative phosphorylation (OXPHOS), fatty acid oxidation, or mitochondrial DNA mutation [5,6,7,8,9].

Mitochondrial dysfunction is also a possible mechanism in DILI due to anti-TB drugs. In mouse experiment, co-administration of isoniazid (INH) and rifampicin (RIF) caused steatosis, increased mitochondrial oxidative stress and necrosis of hepatocytes [10]. In cultured mouse hepatocytes, inhibition of mitochondrial complex I by rotenone precipitated INH-induced cell necrosis, which may be due to simultaneous inhibition of complex I and II (the latter by hydrazine) [11]. Thus, mitochondrial dysfunction, especially those involving complex I, the largest enzyme catalyzing OXOPHOS system, may be involved in DILI by anti-TB drugs. This remains, however, debatable since a recent *in-vivo* study using mouse model reveals contradictory findings, showing that cotreatment with rotenone and INH did not result in histological evidence of liver injury in wild-type mice and mice with impaired immune tolerance [12].

Mitochondrial proteins have two generic origins. The majority of mitochondrial protein, including a major part of complex I and entire complex II, are coded by nuclear DNA. A small part of complex I, and the entire complex III, IV and V (ATP synthase) of the respiratory chain are coded by mitochondrial (mt) DNA [13]. Human mtDNA, a circular DNA consisting of 16,569 base pairs, is located in mitochondrial matrix. Studies have associated mtDNA mutations involving the respiratory chain with mitochondrial diseases [14] and nonalcoholic fatty liver disease [15]. A recent report on patients with biliary atresia described nonsynonymous mtDNA mutations in coding genes of complex I to V that located in critical positions for the proton-pumping activity or enzyme subunit assembly [16]. However, whether mitochondrial DNA variants are associated with DILI due to anti-TB drugs has not been reported. The hypothesis of our study is that mutations in mtDNA could be risk factors of DILI due to isoniazid, rifampicin and pyrazinamide. We employed next-generation sequencing (NGS) to sequence the entire mtDNA genome of patients with and without DILI due to anti-TB drugs and determined the effects of mtDNA variants on the risk of DILI.

## 2. Patients and Methods

### 2.1. Study Design

This prospective study was conducted in National Taiwan University Hospital and according to the Declaration of Helsinki guidelines. All participants gave informed written consent. The hospital’s Institution Review Board approved the study (NTUH REB No.: 9561707008).

The case-control study compared peripheral blood mtDNA variants in patients with and without DILI due to anti-TB drugs. We used peripheral blood mtDNA because a high degree of homology (approximately 98%) has been reported between peripheral blood and liver mitochondrial genomes in patients with non-alcoholic fatty liver disease [15]. We also investigated the *N-*acetyltransferase 2 (*NAT2*) genotype [17].

### 2.2. Patients and Definition of DILI

From March 2007 to December 2010, adult patients with culture-proven tuberculosis (TB) were enrolled at the time of TB diagnosis and followed up prospectively for hepatotoxicity. Individuals were excluded if they were pregnant, had abnormal baseline liver function test (LFT), alcoholism, life expectancy < 6 months, or *Mycobacterium tuberculosis* isolates resistant to rifampicin, isoniazid, or both.

All participants had LFT, including aspartate transaminase (AST), alanine transaminase (ALT), total and direct bilirubin, creatinine and hemogram determined before and every two weeks after starting anti-TB treatment for 2 months, or any time when symptoms/signs of hepatitis developed during anti-TB treatment. HBV, HCV and HIV serology were checked before treatment initiation. *N-*acetyltransferase 2 (*NAT2*) genotype was also determined (see Appendix A for details) [4,18].

All patients received standard anti-TB treatment with daily isoniazid, rifampicin, ethambutol and pyrazinamide for the initial 2 months, and daily isoniazid and rifampicin for the subsequent 4 months with dosages adjusted by weight [19].

DILI was defined as ALT ≥5 times upper limit of normal (ULN), or ALT ≥3 times ULN with total bilirubin ≥2 times ULN [20]. During anti-TB treatment, when DILI occurred, potentially liver-toxic drugs (isoniazid, rifampicin and pyrazinamide) were all discontinued. Re-challenge was commenced when ALT decreased to <2× ULN and total bilirubin <2× ULN. Diagnosis of isoniazid- or rifampicin-induced liver injury required a positive re-challenge test, with doubling of ALT or total bilirubin and recurrence of symptoms after re-challenge. Pyrazinamide-induced liver injury was diagnosed using a positive re-challenge test or by exclusion [4].

A total of 38 patients with TB (men: 20) who had documented DILI consented to participate in the study: 16 with DILI due to pyrazinamide, 14 rifampicin, and 8 isoniazid. We selected 38 sex- and age-matched patients with TB but without DILI as non-DILI controls.

### 2.3. DNA Extraction

Peripheral blood was obtained before the initiation of anti-TB treatment. Total genomic DNA was extracted from leukocytes using a commercial kit (Puregene Blood Core Kit C, QIAGEN, Hilden, Germany).

### 2.4. Sequencing of the Entire Mitochondrial Genome by Using NGS

NGS was performed using PacBio single-molecule real-time (SMRT) sequencing (Pacific Biosciences, CA, USA) [21,22] to detect point mutations in the entire mtDNA. Sequence reads were aligned to the standard revised Cambridge Reference Sequence (rCRS NC_012920.1, Homo sapiens mitochondrion complete genome). Variants were defined as positions with a nucleotide different from reference sequence rCRS (see Appendix A for details). Haplogroup-associated polymorphism was defined as variants which developed during evolution and are related to haplogroups, as analyzed through complete mtDNA sequencing of 560 individuals, and non-haplogroup-associated (private) polymorphism are rare variants that are not associated with evolution or haplogroups, but found only in a small population and not passed to future generations [23].

### 2.5. Statistical Analysis

Quantitative data are shown as mean ± standard deviation and compared using *t*-test. Percentages of cases with a certain point mutation were compared using the z-test or chi-squared test.

## 3. Results

Clinical characteristics of patients with and without DILI are presented in Table 1. No differences were discovered between patients with DILI and those without regarding age, sex, body mass index, smoking history, coexisting diseases, and *NAT2* genotype distribution. Sixteen (42%) patients with DILI and 23 (61%) without DILI received medications other than anti-TB drugs. Five (13%) in the former group and nine (24%) in the latter had received potentially hepatotoxic drugs (*p* = 0.237) (Appendix A). All the hepatotoxic drugs had been prescribed for more than 1 month prior to anti-TB treatment.

ALT level peaked 34 days after initiation of anti-TB treatment, with mean peak ALT 12.5× ULN. Patients with DILI had a 74% anti-TB treatment completion rate at the end of 2 years, significantly lower than that in non-DILI group (95%, *p* = 0.028). Patients with rifampicin-induced liver injury had the lowest treatment completion rate in 2 years (57%, *p* = 0.004 vs. non-DILI group), followed by those with DILI due to isoniazid (63 %, *p* = 0.042) and pyrazinamide (94%). All DILI patients who failed to complete anti-TB treatment died before completion. The 90-day mortality rate was 11% and 5% (*p* = 0.673), and the 2-year mortality rate 26% and 8% (*p* = 0.139) in the DILI and non-DILI group, respectively. Appendix A shows the characteristics of patients who died within 2 years after initiation of anti-TB treatment. The patients in DILI (66 ± 17 years, *p* = 0.280 vs. all DILI cases) and non-DILI (74 ± 19 years, *p* = 0.130 vs. all non-DILI cases) groups who died within 2 years were older than those who survived, but differences were nonsignificant. None of DILI patients required liver transplantation or died of DILI. Patients with rifampicin-induced liver injury had significantly higher 2-year mortality rate (6/14, 43%) than those without DILI (8%, *p* = 0.011). Among the 6 patients with rifampicin-induced liver injury that died within 2 years, the death was directly related to pulmonary TB in 4 (67%).

Comparing the sequences of the patients with the reference sequence, we discovered that in 8 nucleotide positions, the sequences in all of our patients were different from those in the reference (Appendix A). The positions are all haplogroup-associated polymorphisms [23]. We have not counted these nucleotide positions as variants.

In total, we found 1985 variants in 558 mitochondrial positions (Appendix A), 972 in DILI and 1013 in non-DILI group (Table 2). The nucleotide substitution rate is 3.36% (558/16569). In 484 (87%) of the 558 nucleotide positions, the polymorphism had low frequency (<10%) (Figure 1). We observed 3 variants that have not been previously reported: (1) A1390G in *RNR1*, the region that synthesizes 12S ribosomal RNA, in a patient without DILI; (2) C5232A in *ND2*, the region that synthesizes NADH dehydrogenase subunit 2, with proline to threonine substitution, in one patient with and one without DILI; (3) A13930G in *ND5*, the region that synthesizes NADH dehydrogenase subunit 5, with isoleucine to valine change, in a patient without DILI.

Table 2 shows that the average number of mtDNA variants in the entire mitochondrial genomes of patients with (25.6 ± 6.0) and those without DILI (26.7 ± 7.9), were not significantly different (*p* = 0.504). However, in gene *ND5*, there were significantly more mtDNA variants in patients with than in those without DILI (average 2.6 vs. 2.0 variants per person, *p* = 0.028). Further analysis revealed that the significant difference existed between patients with isoniazid-induced liver injury and those without DILI (average 3.5 vs. 2.0 variants per person, *p* = 0.005). Moreover, in gene *ND1* (coding NADH dehydrogenase subunit 1 of complex I), the average number of mtDNA variants in patients with isoniazid-induced liver injury was significantly higher than that of the non-DILI group (2.1 vs. 1.2, *p* = 0.025). However, in patients with rifampicin- or pyrazinamide-induced liver injury, mtDNA variants were not different compared with patients without DILI in any locus of the mitochondrial genome.

Regarding nonsynonymous substitutions that result in amino acid alterations, 192 nonsynonymous substitutions were discovered in DILI and 177 in non-DILI group (Table 3). The average number of nonsynonymous substitutions in each mitochondrial gene was not significantly different between DILI and non-DILI groups, except for *ND5*, for which the average number of nonsynonymous substitutions in patients with isoniazid-induced liver injury was significantly higher than that in the non-DILI group (1.5 ± 1.6 vs. 0.7 ± 0.8, *p* = 0.033). Rifampicin- or pyrazinamide-induced liver injury was not associated with increased number of nonsynonymous substitutions in any mitochondrial gene.

We also observed that for the entire mitochondrial genome, the average ratio of nonsynonymous to all substitutions in DILI group was significantly higher than that in non-DILI group (Table 3). On further analysis, this difference was identified between patients with isoniazid-induced liver injury (but not rifampicin- or pyrazinamide-induced liver injury) and non-DILI group. The ratio of nonsynonymous to synonymous substitutions (Appendix A) was 0.25 (192/780) and 0.21 (177/836) for DILI and non-DILI groups, respectively (*p* = 0.212). In most mitochondrial genes, the ratio was <1, indicating numerous synonymous substitutions. We found that the *COX1* and *ND4* genes had the smallest ratios and the *ATP6*, *ATP8* and *ND6* genes the largest ratios (>1).

The majority of nucleotide substitutions are transitions (purine to purine or pyrimidine to pyrimidine). The percentages of G to A or A to G transition, C to T or T to C transition, and transversions (substitution between a purine and a pyrimidine) were 43.0%, 51.4% and 5.6% in the DILI group, and 44.6%, 47.4% and 8.0% in the non-DILI group, respectively (*p* = 0.084).

We examined the significance of non-haplogroup-associated (“private”) polymorphisms in our study. Among the 558 sites of polymorphisms in our patients, 68 were haplogroup-associated (Appendix A). When we subtracted the polymorphisms in these 68 sites from the total variants and re-analyzed the data, 644 (average 17.0 ± 3.8) and 704 (average 18.5 ± 5.6) private variants were discovered in the DILI and non-DILI groups, respectively (*p* = 0.153, Table 4). The average number of private variants in every coding gene in patients with DILI compared with that in those without DILI was not significantly different except for *ND5*, for which DILI patients had significantly more private variants than non-DILI patients (2.0 ± 1.6 vs. 1.2 ± 1.3, *p* = 0.024). On further analysis, patients with isoniazid-induced liver injury had more private variants than non-DILI patients (2.8 ± 1.8 vs. 1.2 ± 1.3, *p* = 0.007), but not patients with rifampicin- or pyrazinamide-induced liver injury. We also found that in *ND1*, the average number of private variants in patients with isoniazid-induced liver injury was significantly higher than that in non-DILI patients (2.1 ± 1.2 vs. 1.1 ± 1.0, *p* = 0.027). Therefore, analysis on private variants revealed similar results as all variants.

## 4. Discussion

Our study shows that, compared with TB patients without DILI: (1) those with DILI had more mtDNA variants in *ND5* gene; and (2) Patients with isoniazid-induced DILI not only had more mtDNA variants in *ND1* and *ND5* genes but also had more nonsynonymous substitutions, which lead to amino acid changes, in *ND5* gene, and a higher average ratio of nonsynonymous to all substitutions across the entire mitochondrial genome.

Mechanisms of DILI caused by isoniazid, rifampicin and pyrazinamide are not well understood, despite numerous investigations. Mitochondrial dysfunction–including mitochondrial permeability transition pore opening, inhibition of mitochondrial fatty acid oxidation, OXPHOS impairment, and mtDNA damage—has been observed in DILI [5]. The OXPHOS system consists of a series of enzymes, including complex I (NADH: ubiquinone oxidoreductase), II (succinate dehydrogenase), III (ubiquinol-cytochrome c reductase or cytochrome b), IV (cytochrome c oxidase) and V (ATP synthase), all located in the inner membrane of mitochondria, and this system completes the final part of energy generation. The largest enzyme unit of the OXPHOS system is complex I. Complex I comprises three functional parts. The first part (coded by nuclear DNA) is the dehydrogenase, which binds with NADH and oxidizes it to NAD+. The second part is the hydrogenase (also coded by nuclear DNA), which transfers the electrons produced during NADH oxidation to ubiquinone. The third part consists of ND1 to ND6 and ND4L (all coded by mitochondrial DNA), which utilizes the energy released during electron transfer to pump four protons from mitochondria matrix to intermembrane space, creating the proton gradient potential needed for ATP production [24].

Treating rat hepatocytes with hydrazine (a complex II inhibitor) [11] resulted in mega-mitochondrial formation, followed by apoptosis [25,26]. Hydrazine’s effect alone, however, did not cause cell death. However, exposure of mouse hepatocytes to a nontoxic dose of isoniazid and a complex I inhibitor (rotenone or piericidin A) resulted in massive loss of cellular ATP and cell death [11,27].

Compatible with these findings, the results of our study suggest that TB patients with pre-existing *ND5/ND1* nonsynonymous variants which lead to amino acid substitutions might have baseline mitochondrial complex I dysfunction due to proton translocation abnormality (which is explained in the next paragraph). The mild baseline dysfunction alone may not cause overt cell damage because of the large reserve of mitochondria in liver cells. However, when they are treated with isoniazid, their mitochondrial complex II can be inhibited by hydrazine, a metabolite of isoniazid. With simultaneous interference of functions of complex I and complex II, mitochondrial respiratory chain might not be able to supply electrons and translocate protons efficiently, and the resultant ATP shortage might lead to liver cell death. 

Among all mtDNA encoded subunits of complex I, ND5 is the largest and composed of discontinuous helices together with ND4 and DN2, suggesting their function in ion-translocating and proton pumping [28]. Thus nonsynonymous *ND5* polymorphisms with amino acid substitutions may interfere with proton translocation and subsequent ATP production. The smaller ND1 subunit is located at the interface between membrane and matrix arm. The x-ray structure showed that ND1’s amino acids form a narrow entry for ubiquinone to reach its electron donor [29]. *ND1*’s polymorphisms may result in amino acid substitutions and interfere with electron acceptance by ubiquinone. Failure of ubiquinone reduction may interfere with subsequent proton translocation and final ATP production. Thus, the combination of underlying *ND5* and/or *ND1* variants (which interfere with complex I OXOPHOS function), and isoniazid usage with metabolite hydrazine, a complex II inhibitor [11], may lead to liver cell ATP crisis and necrosis. 

In our study, the observed effects of mtDNA polymorphisms were cumulative effects of multiple variants in *ND5* and *ND1*, not the influence of one specific variant, unlike in previous studies of inherited mitochondrial diseases [14]. We did not find any specific variant that predicted increased risk of DILI due to isoniazid. It suggested that most individual variants affected OXPHOS function mildly. The presence of several nonsynonymous variants might augment the effects on OXPHOS and increase the risk of isoniazid-induced liver injury. However, in inherited mitochondrial diseases, individual mutations affect OXPHOS system more critically. For example, In Leber’s inherited optic neuropathy (LHON), the most common mutation is G3460A of *ND1* with alanine substituted by a larger threonine [30] at the narrow ubiquinone entrance and interfering with ubiquinone reduction [29].

In their analysis of 560 mitochondrial DNA coding region sequences, Herrnstadt el al. [23] discovered that 497 polymorphisms were haplogroup-associated. These polymorphisms are related to evolution and occur in different combination groups during evolution. Different ethnic groups have characteristics haplogroups. Some haplogroup-associated polymorphisms are risk factors of ischemic or hypertrophic cardiomyopathy [31,32]. Regarding private polymorphisms, the association with diseases is not well established. Our results demonstrated an association between *ND1*/*ND5* variants and isoniazid-induced liver injury for all as well as for only private variants. 

Our study did not find association between mitochondrial polymorphism and rifampicin- or pyrazinamide-induced liver injury. Mechanisms of rifampicin-induced liver injury are not clear, but rifampicin caused lipid accumulation and increased oxidative stress in rat hepatocytes [33]. Mechanisms of pyrazinamide-induced liver injury have been investigated in a zebra fish larvae model, and peroxisome proliferator-activated receptor α (PPAR-α) downregulation with enhanced oxidative stress were associated with pyrazinamide-induced liver injury [34]. Whether these findings are related to mitochondrial dysfunction is uncertain.

Our study has some limitations. First, many of our patients had coexisting diseases and were receiving treatment for these diseases. Although we selected a matched non-DILI group, the two groups may still have different backgrounds, which could confound study results. Second, we did not measure serum concentration of anti-TB drugs, which could make the diagnoses of causative drug of liver injury less definite. Third, we used peripheral blood leukocyte rather than liver as the source of mtDNA. Although peripheral blood and liver mitochondrial genomes in patients with non-alcoholic fatty liver disease were shown to have a high degree homology (approximately 98%) [15], the extent of heteroplasmy in patients with DILI is unknown. Liver biopsy is, however, not indicated for DILI during anti-TB treatment and is seldomly performed because the diagnosis is often possible based on the known hepatotoxic potential and temporal compatibilities. Fourth, the sample size was small.

## 5. Conclusions

Our study discovered that TB patients with isoniazid-induced liver injury had more mitochondrial DNA variants in *ND5* and *ND1* gene that code part of complex I, more nonsynonymous substitutions in *ND5*, and a higher ratio of nonsynonymous to all substitutions across the entire mitochondrial genome. The variants might affect the OXPHOS function of complex I, and exposure of these patients to isoniazid’s metabolite hydrazine (a complex II inhibitor) might predispose them to OXPHOS failure and isoniazid-induced liver injury.

## Figures and Tables

**Figure 1 jcm-08-01207-f001:**
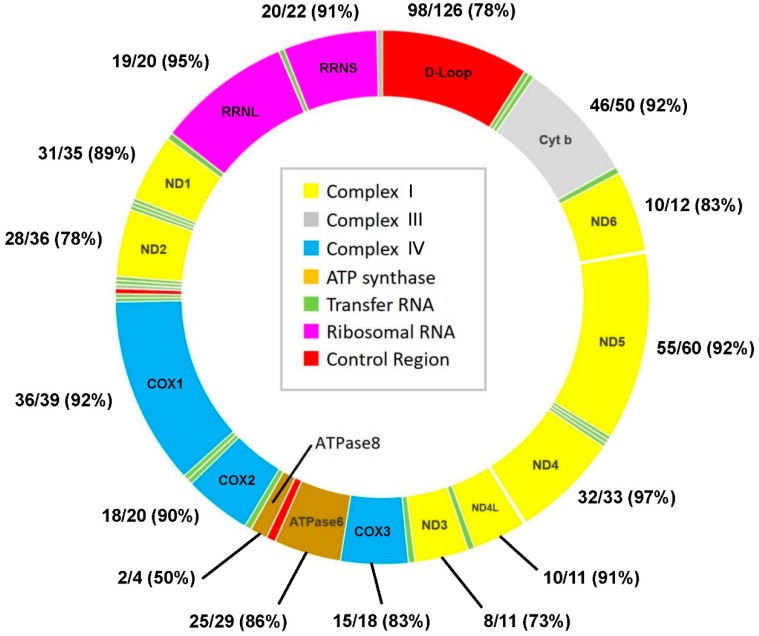
The map of human mitochondrial DNA, and number and percentage of low-frequency polymorphisms among all polymorphisms in each region. (There were 28 polymorphisms in the tRNA region, and all were low-frequency polymorphisms. Three of the four polymorphisms in the intron region were low-frequency polymorphisms.)

**Table 1 jcm-08-01207-t001:** Clinical characteristics and *N-*acetyltransferase 2 (*NAT2*) genotype of tuberculosis (TB) patients with and without drug-induced liver injury (DILI) due to anti-TB drug.

Variable	Patients with DILI (*n* = 38)	Patients without DILI (*n* = 38)	*p*
Age (years)	58.7 ± 18.9	55.5 ± 20.0	0.476
Female	18 (47)	18 (47)	0.818
Body-mass index (kg/M2)	20.9 ± 3.26	20.7 ± 2.80	0.881
Never smoker	21 (55)	23 (61)	0.821
Co-existing diseases			
Diabetes mellitus	6 (16)	8 (21)	0.764
Malignancy	6 (16)	8 (21)	0.764
Heart failure	4 (11)	2 (5)	0.678
CKD, stage 4	2 (5)	0	0.469
CKD, post-renal transplant	0	2 (5)	0.469
Autoimmune disease	3 (8)	1 (3)	0.602
*NAT2* genotype*: *n*/total (%)			
Slow acetylator	13/34 (38)	12/38 (32)	0.835
Intermediate acetylator	11/34 (32)	14/38 (37)	
Rapid acetylator	10/34 (29)	12/38 (32)	
Slow and intermediate acetylator	24/34 (70)	26/38 (68)	0.948
DILI due to INH	8/8 (1)		0.154
Days to peak ALT: M ± SD (Range)	33.8 ± 23.6 (2–117)	38.8 ± 27.4 (4–92)	0.398
Peak ALT value (ULN): M ± SD (Range)	12.5 ± 9.7 (3–47.3)	1.33 ± 0.99 (0.3–4)	0.000
Complete anti-TB Tx in 2 years: *n*/total (%)	28/38 (74)	36/38 (95)	0.028
In patients with DILI due to INH	5/8 (63)		0.042
In patients with DILI due to RIF	8/14 (57)		0.004
In patients with DILI due to PZA	15/16 (94)		0.604
90-day mortality rate: *n*/total (%)	4/38 (11)	2/38 (5)	0.673
In patients with DILI due to INH	1/8 (13)		0.970
In patients with DILI due to RIF	2/14 (14)		0.622
In patients with DILI due to PZA	1/16 (6)		0.610
2-year mortality rate: *n*/total (%)	10^#^/38 (26)	3^§^/38 (8)	0.068
In patients with DILI due to INH	3/8 (38)		0.093
In patients with DILI due to RIF	6/14 (43)		0.011
In patients with DILI due to PZA	1/16 (6)		0.716

Abbreviations: ALT, alanine transaminase; CKD, chronic kidney disease; INH, isoniazid; M ± SD, mean ± standard deviation; PZA, pyrazinamide; RIF, rifampin; TB, tuberculosis; Tx, treatment; ULN, upper limit of normal. Data are either number (%) or mean ± SD, unless otherwise mentioned. * *NAT2* genotype was available in 34 patients with DILI and 38 patients without DILI. ^#^ All died before completion of anti-TB treatment. ^§^ Two died before completion of anti-TB treatment.

**Table 2 jcm-08-01207-t002:** Total and average number of variants per patient in each mitochondrial mtDNA gene for TB patients with DILI (*n* = 38) and without DILI (*n* = 38).

mtDNA Region/Gene	Total and Average No. of Variants Per Patient		*p*
	DILI (*n* = 38)	No DILI (*n* = 38)	
D-loop	251 (6.6 ± 2.4)	289 (7.6 ± 2.7)	0.094
RRNS (12S ribosomal RNA)	26 (0.7 ± 0.8)	36 (1.0 ± 0.8)	0.142
RRNL (16S ribosomal RNA)	26 (0.7 ± 0.9)	15 (0.4 ± 0.6)	0.103
ND1 (NADH dehydrogenase subunit 1)	54 (1.4 ± 1.1)	46 (1.2 ± 1.0)	0.425
DILI due to INH (*n* = 8)	17 (2.1 ± 1.2)		0.025
DILI due to PZA (*n* = 16)	17 (1.2 ± 1.3)		0.947
DILI due to RIF (*n* = 14)	20 (1.4 ± 0.8)		0.447
ND2 (NADH dehydrogenase subunit 2)	89 (2.3±1.2)	94 (2.5 ± 1.3)	0.644
DILI due to INH (*n* = 8)	16 (2.0 ± 1.1)		0.331
DILI due to PZA (*n* = 16)	42 (2.6 ± 1.3)		0.684
DILI due to RIF (*n* = 14)	31 (2.2 ± 1.2)		0.502
COX1 (Cytochrome c oxidase subunit 1)	44 (1.2 ± 1)	44 (1.2 ± 1.1)	>0.999
COX2 (Cytochrome c oxidase subunit 2)	18 (0.5 ± 0.7)	23 (0.6 ± 0.9)	0.462
ATP8 (ATP synthase F0 subunit 8)	11 (0.3 ± 0.6)	5 (0.1 ± 0.3)	0.169
ATP6 (ATP synthase F0 subunit 6)	48 (1.3 ± 0.9)	50 (1.3 ± 1.1)	0.815
COX3 (Cytochrome c oxidase subunit 3)	36 (0.9 ± 0.9)	34 (0.8 ± 1.0)	0.524
ND3 (NADH dehydrogenase subunit 3)	52 (1.4 ± 0.9)	51 (1.3 ± 1.0)	0.903
ND4L (NADH dehydrogenase subunit 4L)	16 (0.4 ± 0.6)	10 (0.3 ± 0.6)	0.288
ND4 (NADH dehydrogenase subunit 4)	47 (1.2 ± 1.2)	40 (1.1 ± 1.0)	0.427
ND5 (NADH dehydrogenase subunit 5)	102 (2.6 ± 1.6)	77 (2.0 ± 1.2)	0.028
DILI due to INH (*n* = 8)	28 (3.5 ± 1.6)		0.005
DILI due to PZA (*n* = 16)	40 (2.6 ± 1.5)		0.128
DILI due to RIF (*n* = 14)	39 (2.4 ± 1.6)		0.332
ND6 (NADH dehydrogenase subunit 6) (light strand)	12 (0.3 ± 0.5)	17 (0.4 ± 0.6)	0.292
CYTB (Cytochrome b)	124 (3.3 ± 1.9)	114 (3 ± 1.7)	0.528
tRNA genes	14 (0.3 ± 0.7)	19 (0.5 ± 0.7623)	0.353
Introns	2 (0.1 ± 0.2)	13 (0.3 ± 0.5)	0.001
All variants	972 (25.6 ± 6.0)	1013 (26.7 ± 7.9)	0.504

Abbreviations: DILI, drug-induced liver injury; INH, isoniazid, NADH, nicotinamide adenine dinucleotide; PZA, pyrazinamide; RIF, rifampin; TB, tuberculosis. Data are total number (average number ± standard deviation), unless otherwise mentioned.

**Table 3 jcm-08-01207-t003:** Total and average number of nonsynonymous substitutions in mtDNA genes in TB patients with and without DILI.

mtDNA Gene	Total and Average No. (M ± SD) of NS Substitutions		*p*
	DILI (*n* = 38)	No DILI (*n* = 38)	
ND1 (NADH dehydrogenase subunit 1)	16 (0.42 ± 0.60)	16 (0.42 ± 0.55)	>0.999
ND2 (NADH dehydrogenase subunit 2)	23 (0.60 ± 0.68)	27 (0.71 ± 0.84)	0.550
COX1 (Cytochrome c oxidase subunit 1)	1 (0.03 ± 0.16)	4 (0.11 ± 0.31)	0.169
COX2 (Cytochrome c oxidase subunit 2)	4 (0.11 ± 0.31)	7 (0.18 ± 0.39)	0.335
ATP8 (ATP synthase F0 subunit 8)	6 (0.16 ± 0.37)	5 (0.13 ± 0.34)	0.749
ATP6 (ATP synthase F0 subunit 6)	37 (0.97 ± 0.75)	34 (0.90 ± 0.73)	0.640
COX3 (Cytochrome c oxidase subunit 3)	4 (0.11 ± 0.31)	1 (0.03 ± 0.16)	0.170
ND3 (NADH dehydrogenase subunit 3)	19 (0.5 ± 0.51)	22 (0.58 ± 0.55)	0.520
ND4L (NADH dehydrogenase subunit 4L)	5 (0.13 ± 0.34)	3 (0.08 ± 0.36)	0.515
ND4 (NADH dehydrogenase subunit 4)	4 (0.11 ± 0.31)	3 (0.08 ± 0.36)	0.734
ND5 (NADH dehydrogenase subunit 5)	39 (1.03 ± 1.26)	25 (0.66 ± 0.81)	0.135
DILI due to isoniazid (*n* = 8)	12 (1.50 ± 1.60)		0.033
ND6 (NADH dehydrogenase subunit 6)	7 (0.18 ± 0.39)	9 (0.26 ± 0.45)	0.415
CYTB (Cytochrome b)	27 (0.71 ± 0.77)	21 (0.55 ± 0.69)	0.347
Total No. of NS substitutions	192 (5.39 ± 1.73)	177 (5.24 ± 2.09)	0.735
Average ratio of NS substitutions/all variants	0.20 ± 0.06	0.17 ± 0.06	0.045
DILI due to isoniazid (*n* = 8)	0.24 ± 0.07		0.010
DILI due to pyrazinamide (*n* = 16)	0.20 ± 0.07		0.230
DILI due to rifampin (*n* = 14)	0.18 ± 0.05		0.595

Abbreviations: ATP, adenosine triphosphate; DILI, drug-induced liver injury; NADH, nicotinamide adenine dinucleotide; NS, nonsynonymous; TB, tuberculosis.

**Table 4 jcm-08-01207-t004:** Average number of private variants (with haplogroup-associated variants subtracted from total variants) in mtDNA genes for TB patients with DILI (*n* = 38) and without DILI (*n* = 38).

mtDNA Region/Gene	Total and Average No. of Variants Per Patient		*p*
	DILI (*n* = 38)	No DILI (*n* = 38)	
D-loop	3.7 ± 2.4	4.6 ± 2.2	0.063
RRNS (12S ribosomal RNA)	0.5 ± 0.7	0.6 ± 0.9	0.557
RRNL (16S ribosomal RNA)	0.4 ± 0.7	0.3 ± 0.5	0.186
ND1 (NADH dehydrogenase subunit 1)	1.3 ± 1.1	1.1 ± 1.0	0.236
DILI due to INH (*n* = 8)	2.1 ± 1.2		0.027
DILI due to PZA (*n* = 16)	1 ± 1.1		0.864
DILI due to RIF (*n* = 14)	1.3 ± 0.8		0.444
ND2 (NADH dehydrogenase subunit 2)	1.2 ± 1.2	1.3 ± 1.2	0.704
DILI due to INH (*n* = 8)	0.8 ± 0.7		0.268
DILI due to PZA (*n* = 16)	1.6 ± 1.3		0.341
DILI due to RIF (*n* = 14)	0.9 ± 1.0		0.371
COX1 (Cytochrome c oxidase subunit 1)	1.1 ± 0.9	1.1 ± 1.1	0.825
COX2 (Cytochrome c oxidase subunit 2)	0.4 ± 0.6	0.6 ± 0.9	0.292
ATP8 (ATP synthase F0 subunit 8)	0.1 ± 0.4	0.1 ± 0.2	0.703
ATP6 (ATP synthase F0 subunit 6)	1.3 ± 0.9	1.5 ± 1.1	0.252
COX3 (Cytochrome c oxidase subunit 3)	0.1 ± 0.3	0.03 ± 0.2	0.311
ND3 (NADH dehydrogenase subunit 3)	0.3 ± 0.5	0.4 ± 0.6	0.542
ND4L (NADH dehydrogenase subunit 4L)	0.4 ± 0.5	0.2 ± 0.5	0.202
ND4 (NADH dehydrogenase subunit 4)	0.7 ± 1.0	0.6 ± 0.8	0.902
ND5 (NADH dehydrogenase subunit 5)	2.0 ± 1.6	1.2 ± 1.3	0.024
DILI due to INH (*n* = 8)	2.8 ± 1.8		0.007
DILI due to PZA (*n* = 16)	1.7 ± 1.6		0.235
DILI due to RIF (*n* = 14)	1.9 ± 1.6		0.129
NS substitutions	0.9 ± 1.2	0.5 ± 0.8	0.073
DILI due to INH (*n* = 8)	1.5 ± 1.6		0.011
DILI due to PZA (*n* = 16)	0.7 ± 1.1		0.534
DILI due to RIF (*n* = 14)	0.9 ± 1.1		0.138
ND6 (NADH dehydrogenase subunit 6) (light strand)	0.2 ± 0.4	0.4 ± 0.5	0.534
CYTB (Cytochrome b)	0.7 ± 0.8	0.8 ± 1.1	0.811
All private variants: total (mean ± SD)	644 (17.0 ± 3.8)	704 (18.5 ± 5.6)	0.153

Abbreviations: DILI, drug-induced liver injury; INH, isoniazid, NS, non-synonymous; PZA, pyrazinamide; RIF, rifampin; TB, tuberculosis. Data are total number (average number ± standard deviation), unless otherwise mentioned.

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
