# Peer review of "Mitochondrial DNA Variants in Patients with Liver Injury Due to Anti-Tuberculosis Drugs"

_jcm, 2019, doi:10.3390/jcm8081207_

Round 1

Reviewer 1 Report

Dr. Lee et al. conducted a comprehensive analysis of a mitochondrial DNA variants that affect drug induced liver injury in patients undergoing tuberculosis treatment. They found mutations in several genes, most prominently in sub-unit 1 and 5 of complex I might possibly be a risk factor. However, the following concerns should be raised.

1) ND1 and ND5 expression in human liver: Are there any previous findings to demonstrate that detect differential expression in human liver and/or its correlation with several synonymous/non-synonymous mutations observed here or elsewhere? Information about this might be relevant in the introduction

2) Private Mutations: I believe in general since the article would be of immense interest to general readers as well as researchers/medical practitioners. The authors should therefore do a better job in explaining this term to the readers.

Author Response

1. ND1 and ND5 expression in human liver: Are there any previous findings to demonstrate that detect differential expression in human liver and/or its correlation with several synonymous/non-synonymous mutations observed here or elsewhere? Information about this might be relevant in the introduction

ANS: Thank you for the nice suggestion. A recent report on patients with biliary atresia described nonsynonymous mtDNA mutations in coding genes of complex 1 to V that located in critical positions for the proton-pumping activity or enzyme subunit assembly (Reference 16 of the revised manuscript). We have added the discussion and reference in the last paragraph of the introduction section. Thank you.

“A recent report on patients with biliary atresia described nonsynonymous mtDNA mutations in coding genes of complex 1 to V that located in critical positions for the proton-pumping activity or enzyme subunit assembly [16].”

2. Private Mutations: I believe in general since the article would be of immense interest to general readers as well as researchers/medical practitioners. The authors should therefore do a better job in explaining this term to the readers.

Ans: Thanks for the important reminder. We have defined “private mutation” in the methodology in the revised manuscript (2nd last paragraph of the method section).

“Haplogroup-associated polymorphism was defined as variants which developed during evolution and are related to haplogroups, as analyzed through complete mDNA sequencing of 560 individuals, and non-haplogroup-associated (private) polymorphisms are rare variants that are not associated with evolution or haplogroups, but found only in a small population and not passed to future generations [23].

Reviewer 2 Report

This manuscript described a very interesting association of mitochondrial DNA variants in patients with liver injury due to anti-TB drugs. The authors sequenced the genome of mitochondrial DNA in peripheral blood from TB patients, and found that DILI induced by isoniazid had more variants in complex I’s NADH subunit 5 and 1 genes, more nonsynonymous mutations in NADH subunit 5, and a higher ratio of nonsynonymous to total substitutions. Such kind of gene variants is specially to isoniazid, because DILI induced by other anti-TB drugs rifampicin or pyrazinamide had no association with mitochondrial DNA variants. Although the patient sample size was small, it will provide a significant clinical diagnosis for isoniazid DILI. Some questions may be considered which can provide better evidence in this field.

1.      Although the author mentioned the limitation in the manuscript that peripheral blood DNA, other than liver DNA, was used, is it possible to get some liver sample for sequencing? This will provide much more solid evidence.

2.      Coexisting diseases in these patients may hide some information. Can the author provide the information of cotreatment drugs besides anti-TB drugs in these patients? Whether these drugs have some effect on mitochondrial toxicity or themselves can cause DILI?

3.      A recent paper with the title “Rotenone Increases Isoniazid Toxicity but Does Not Cause Significant Liver Injury: Implications for the Hypothesis that Inhibition of the Mitochondrial Electron Transport Chain Is a Common Mechanism of Idiosyncratic Drug-Induced Liver Injury” published in Chem. Res. Toxicol may be cited. And more discussion should be included in the manuscript since this article mentioned that rotenone, a complex I inhibitor, do not increase isoniazid toxicity in wild type mice.

Author Response

1. Although the author mentioned the limitation in the manuscript that peripheral blood DNA, other than liver DNA, was used, is it possible to get some liver sample for sequencing? This will provide much more solid evidence.

Ans: Thank you for the important comment. We totally agree with the reviewer that heteroplasmy is an important factor in studying mitochondrial dysfunction and disease. Liver biopsy is, however, not indicated for drug-induced liver injury (DILI) during anti-TB treatment and is seldomly performed because the diagnosis of DILI is often possible based on the known hepatotoxic potential and temporal compatibilities. Therefore, we are very sorry that liver sample was not collected in any of the 38 patients with DILI. We have revised the discussion of the limitation to better describe such consideration.

“Third, we used peripheral blood leukocyte rather than liver as the source of mtDNA. Although peripheral blood and liver mitochondrial genomes in patients with non-alcoholic fatty liver disease were shown to have a high degree homology (approximately 98%) [15], the extent of heteroplasmy in patients with DILI is unknown. Liver biopsy is, however, not indicated for DILI during anti-TB treatment and is seldomly performed because the diagnosis is often possible based on the known hepatotoxic potential and temporal compatibilities.”

2. Coexisting diseases in these patients may hide some information. Can the author provide the information of cotreatment drugs besides anti-TB drugs in these patients? Whether these drugs have some effect on mitochondrial toxicity or themselves can cause DILI?

Ans: Thank you for the instructive comment. We have added a new table in the supplementary file (Table S1) and some descriptions in the first paragraph of the results section to summarize the cotreatment drugs.

“Sixteen (42%) patients with DILI and 23 (61%) without DILI received medications other than anti-TB drugs. Five (13%) in the former group and nine (24%) in the latter had received potentially hepatotoxic drugs (p=0.237) (Table S1). All the hepatotoxic drugs had been prescribed for more than 1 month prior to anti-TB treatment.”

Though many of these potentially hepatotoxic drugs have been reported to cause mitochondria dysfunction, all had been prescribed for more than 1 month prior to anti-TB treatment and the proportion of patients receiving these drugs was similar in the two groups. Therefore, it is less likely that these hepatotoxic drugs played a significant role in causing DILI in our patients.

3. A recent paper with the title “Rotenone Increases Isoniazid Toxicity but Does Not Cause Significant Liver Injury: Implications for the Hypothesis that Inhibition of the Mitochondrial Electron Transport Chain Is a Common Mechanism of Idiosyncratic Drug-Induced Liver Injury” published in Chem. Res. Toxicol may be cited. And more discussion should be included in the manuscript since this article mentioned that rotenone, a complex I inhibitor, do not increase isoniazid toxicity in wild type mice.

Ans: Thank you for the nice suggestion. This paper has been cited and discussed in the 2nd paragraph of the introduction section.

“This remains, however, debatable since a recent in-vivo study using mouse model reveals contradictory findings, showing that cotreatment with rotenone and INH did not result in histological evidence of liver injury in wild-type mice and mice with impaired immune tolerance [12].”